# Vulnerability Analysis of Safe Reinforcement Learning via Inverse Constrained Reinforcement Learning

## Abstract

Safe reinforcement learning (Safe RL) aims to ensure policy performance while satisfying safety constraints. However, most existing Safe RL methods assume benign environments, making them vulnerable to adversarial perturbations commonly encountered in real-world settings. In addition, existing gradient-based adversarial attacks typically require access to the policy's gradient information, which is often impractical in real-world scenarios. To address these challenges, we propose a vulnerability analysis framework for Safe RL policies via inverse constrained reinforcement learning (ICRL). Our approach only requires a set of expert demonstrations to learn both the safety constraints and a learner policy, which are then used to generate adversarial attacks capable of inducing safety violations in Safe RL policies. Theoretical analysis establishes the feasibility and provides bounds for our attack method. Experiments on multiple Safe RL benchmarks demonstrate the effectiveness of our approach.

## 1 Introduction

Safe reinforcement learning (Safe RL) integrates safety constraints into policy optimization, enabling agents to balance performance with safety requirements (Gu et al., 2022). Owing to its ability to enforce safety during training and deployment, Safe RL has been widely applied in safety-critical domains, including autonomous driving and robotic manipulation.

However, most existing Safe RL methods assume benign, attack-free environments, leaving them vulnerable to adversarial perturbations that frequently occur in the real world. Recent studies have investigated this vulnerability by designing adversarial attacks targeting the observation space of trained Safe RL agents (Liu et al., 2022b; Jiang et al., 2024). While insightful, these approaches typically rely on strong assumptions such as access to the policy's internal gradients (Liu et al., 2022b) or knowledge of the ground-truth safety constraints (Jiang et al., 2024), which limits their applicability in practical attack scenarios.

These limitations raise a fundamental question: *Can we systematically reveal and evaluate the vulnerabilities of Safe RL policies in a general and practical setting that relies solely on access to expert demonstrations?* Addressing this problem poses several key challenges. First, designing effective adversarial perturbations often requires access to privileged information, such as the policy's internal gradients or safety constraints, which may not be realistic in practice. Second, how can we ensure that adversarial inputs result in actual safety violations instead of simply lowering performance or rewards? Third, how can we determine an attack strength that leads to safety violations without compromising stealth?

To tackle these challenges, we propose an adversarial attack framework based on inverse constrained reinforcement learning (ICRL) for Safe RL policies. Instead of using the policy's internal gradients or safety constraints to generate adversarial perturbations, our method leverages ICRL to learn the safety constraints and a learner policy from expert demonstrations. Then, the learned constraints and learner policy are used to generate adversarial perturbations that cause higher cost values of safety constraint violations. Also, the learned constraints provide a theoretical bound on perturbations to estimate the optimal attack strength.

In summary, our contributions are as follows:

1. We propose a practical and effective vulnerability analysis framework for Safe RL. Our adversarial attack-based framework relies solely on expert demonstrations to generate effective perturbations that lead to safety constraint violations, providing a practical and generalizable tool for Safe RL vulnerability analysis and offering new insights into the weaknesses of Safe RL systems.

2. We provide theoretical analyses of the proposed framework. We theoretically verify its feasibility, local optimality, and derive an upper bound on the attack effectiveness.

3. We demonstrate the effectiveness of our method on multiple Safe RL environments. The results show that our method generalizes across unseen expert policies and environments while exposing vulnerabilities in well-trained Safe RL agents.

## 2 RELATED WORK

**Safe Reinforcement Learning (Safe RL).** Safe RL aims to ensure policy performance while satisfying safety constraints, which are increasingly applied in safety-critical applications, such as autonomous driving (Wachi et al., 2024) and CPS (Bui et al., 2024). Lagrange-based Safe RL methods (Ray et al., 2019; Stooke et al., 2020) formulate safety constraints as penalties in the reward function using Lagrange multipliers to balance reward maximization and constraint satisfaction. Ji et al. (2024) developed a comprehensive Safe RL benchmark, OmniSafe, which includes a suite of Safe RL algorithms, a Gymnasium-like API, and a diverse set of safety-critical environments. Brunke et al. (2022) provides a systematic review of Safe RL methods in robotics, discussing different levels of safety and summarizing the corresponding technical approaches.

**Adversarial attacks on RL** Goodfellow et al. (2014) first proposed the gradient-based adversarial attack method, fast gradient sign method (FGSM), for neural network-based policies. Further, Huang et al. (2017) extends FGSM to RL algorithms. Additionally, defense methods have been proposed to enhance policy robustness. For instance, Liang et al. (2022) introduces a robust training framework to optimize reward when the policy is under adversarial attack. Zhang et al. (2020) introduces state-adversary Markov Decision Process (SA-MDP) for robustness improvement of RL algorithms.

There are also some works on adversarial attacks on Safe RL policies. Liu et al. (2022b) proposed an approach targeting Safe RL policies to manipulate the gradient of the reward critic and safety critic to violate safety constraints. However, this method requires the policy's gradient information. Jiang et al. (2024) introduced an adversarial attack method on Safe RL systems, which is guided by robustness values derived from the quantitative semantics of STL. However, it assumes prior knowledge of the system's safety constraints.

**Inverse Constrained Reinforcement Learning (ICRL).** Scobee & Sastry (2019) extended MaxEnt IRL method to infer safety constraints from constrained RL settings. Further, Malik et al. (2021) proposed a learning-based ICRL method that infers safety constraints using a neural network classifier. Liu et al. (2022a) introduced an ICRL benchmark. Kim et al. (2024) proposed an algorithm for learning shared safety constraints across multi-task RL environments. McPherson et al. (2021) introduced an extension designed for stochastic environments. Lindner et al. (2024) proposed an ICRL method that learns a function that infers both rewards and constraints simultaneously. Qiao et al. (2024) introduced a multi-modal ICRL approach capable of distinguishing different agent trajectories and learning multiple sets of constraints for various behaviors. Additionally, Hugessen et al. (2024) proposed a method that reduces the ICRL problem to a standard IRL problem. Lastly, Xu & Liu developed a robust ICRL method that accounts for real-world variations, mitigating the assumption that training and test environments are identical—an unrealistic condition.

## 3 PROBLEM FORMULATION

### 3.1 SAFE REINFORCEMENT LEARNING

Safe RL integrates safety constraints into the policy learning process, ensuring that the policy not only maximizes expected rewards but also adheres to predefined safety requirements. The

problem is typically formulated as a Constrained Markov Decision Process (CMDP), represented as $(S, A, r, c, p, \gamma)$, where $S$ is the state space; $A$ is the action space; $r : S \times A \times S \to \mathbb{R}$ denotes the reward function; $c : S \times A \times S \to \mathbb{R}^+$ denotes the cost function for safety constraint violations; $p : S \times A \times S \to [0, 1]$ is the transition probability function; and $\gamma \in [0, 1)$ is the discount factor. The objective of Safe RL is to maximize the expected cumulative reward while ensuring that the expected cost remains below a predefined threshold $d$, formally expressed as:

$$\pi^* = \arg\max_{\pi} V_r^{\pi}, \quad \text{s.t.} \quad V_c^{\pi} \leq d, \tag{1}$$

where $V_r^{\pi}$ denotes the expected cumulative reward under policy $\pi$, and $V_c^{\pi}$ denotes the expected cumulative cost. Various algorithms extend traditional RL methods to address such constrained optimization problems, including PPO-Lagrangian (Ray et al., 2019) and PID-Lagrangian-TRPO (Stooke et al., 2020).

## 3.2 THREAT MODEL

**Attack Setup:** We consider adversarial attacks on an expert policy $\pi_E$ trained by a safe RL algorithm. At each time step $t$, the attacker adds a perturbation $\delta_t$ to the agent's observation $s_t$, yielding a modified input $\hat{s}_t = s_t + \delta_t$. The expert policy then selects an action $\hat{a}_t = \pi_E(\hat{s}_t)$, which drives the environment transition $s_{t+1} = p(s_{t+1} \mid s_t, \hat{a}_t)$. The attacker applies such perturbations throughout the episode.

**Attacker's Capability:**

- The attacker can perturb observations $s_t$ within a norm-bounded region: $\|\delta_t\| \leq \epsilon$, where $\epsilon$ is a user-defined attack budget.

- The attacker does not modify the environment dynamics, rewards, or constraints directly.

**Adversarial Knowledge:** The attacker operates under a minimal assumption of the victim's knowledge:

- The attacker has **no access** to the gradient information of $\pi_E$.

- The attacker does **not know** the environment's true reward or constraint functions.

- The attacker can collect a **limited number of expert trajectories** $\mathcal{D}_E = \{\tau_i\}_{i=1}^N$, where each trajectory $\tau_i = \{(s_t, a_t, r_t, c_t)\}_{t=0}^T$.

## 3.3 ATTACK METRICS

We make the following definitions and assumptions for the adversarial attack on Safe RL.

**Attack Effectiveness:** We define an adversarial attack on the expert policy $\pi_E$ as effective if introducing a perturbation $\delta$ increases the constraint violation cost relative to the original (unperturbed) scenario. Formally, this effectiveness condition can be expressed as: $V_c^{\pi_E^{\psi}}(s + \delta) > V_c^{\pi_E^{\psi}}(s)$, where $V_c^{\pi_E^{\psi}}(s + \delta)$ represents the cumulative constraint violation cost when the policy input state $s$ is perturbed by $\delta$.

**Attack Strength:** The strength of an adversarial attack is characterized by the magnitude of the perturbation applied to the input state. It is quantified by the $L_\infty$-norm of the difference between the original state $s$ and the perturbed state $\tilde{s}$, constrained by the attack budget $\epsilon$: $\|\tilde{s} - s\|_\infty \leq \epsilon$, where $\epsilon$ controls the maximum allowable deviation and thus determines the strength and subtlety of the attack.

**Attack Stealthiness:** We define stealthiness as the ability of an attack to succeed under limited knowledge assumptions. In practice, more stealthy attacks retrieve less information from the victim system, which makes them harder to detect, whereas less stealthy ones require stronger access such as gradients or safety constraints. Table 1 summarizes the knowledge access levels.

Table 1: Knowledge access levels for adversarial attacks on Safe RL. Lower levels correspond to weaker assumptions and higher stealthiness.

| Level | Knowledge Required | Stealthiness |
|-------|--------------------|--------------|
| L1 | Expert trajectories only | High |
| L2 | Trajectories + partial extra knowledge (safety constraints *or* policy gradients) | Medium |
| L3 | Trajectories + full knowledge (safety constraints + policy gradients, etc.) | Low |

## 4 METHOD

In this section, we first provide definitions and assumptions regarding adversarial attacks in Safe RL. Then, we introduce our proposed adversarial attack method based on ICRL. An overview of the proposed method is shown in Figure 1.

### 4.1 LEARNING CONSTRAINTS VIA ICRL

The inverse constrained reinforcement learning (ICRL) (Kim et al., 2024) assumes that the safety constraints of the environment are unobservable or difficult to formulate mathematically. It aims to learn the safety constraint function $\psi$ from expert demonstrations. One representative ICRL method is using contrastive learning to learn a neural network discriminator by distinguishing the expert demonstrations from the learner demonstrations. It consists of two steps: constrained RL policy (learner policy) learning and safety constraint learning. The learner policy learning aims to learn a policy that has task reward as close as the expert policy, as well as keeping the costs of safety violation no more than the expert policy. It can be formulated as a min-max problem:

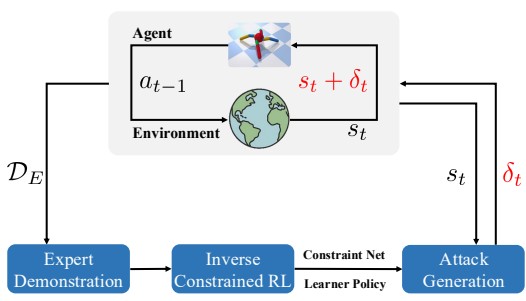

Figure 1: An overview of the proposed adversarial attack framework.

$$\min_{\pi_L \in \Pi} V_r^{\pi_E} - V_r^{\pi_L}$$
$$\text{s.t. } \max_{\psi \in \mathcal{F}_\psi} \left( V_c^{\pi_L^\psi} - V_c^{\pi_E^\psi} \right) \leq 0, \tag{2}$$

where $\pi_L$ represents learner policy; $V_r^{\pi_L}$ and $V_r^{\pi_E}$ are the expected cumulative reward under $\pi_L$ and $\pi_E$; $\mathcal{F}_\psi$ is the set of all possible constraints; $V_c^{\pi_L^\psi}$ is the expected cumulative cost under $\pi_L$ and constraint function $\psi$; $V_c^{\pi_E^\psi}$ is the cumulative cost of the expert policy $\pi_E$ under the learned constraint $\psi$. The second step is to learn the safety constraint function $\psi$ from expert demonstrations by identifying behavioral differences between the expert and the learner. To do so, we optimize for a constraint function that maximally separates the expert policy $\pi_E$ from the set of previously learned learner policies $\{\pi_{Lj}\}_{j=1}^T$. The optimization objective is formulated as

$$\psi \leftarrow \arg\max_{\psi \in \mathcal{F}_\psi} \sum_{j=1}^{T} \left[ V_c^{\pi_{Lj}^\psi} - V_c^{\pi_E^\psi} \right] - R(\psi), \tag{3}$$

where $V_c^{\pi_{Lj}^\psi}$ is the expected cumulative cost under policy $\pi_{Lj}^\psi$; $R(\psi)$ is the regularization term, instantiated as the $L_2$-norm in practice to prevent overfitting. Equation 3 encourages the learned constraint to penalize the learner's behavior more than the expert's, capturing implicit constraints obeyed by the expert demonstrations.

Finally, we can get a constraint function $\psi$ and a learner policy $\pi_L$ from the ICRL process.

---

**Algorithm 1:** Attack Generation

---

**Data:** Learner policy $\pi$, system identification function $\hat{f}$, constraint function $\psi$, expert policy's observation $s_t$, perturbation bound $\epsilon$, number of iterations $n$, step size $\kappa$

**Result:** Perturbation $\delta_t$

1 $s'_t \leftarrow s_t$;           // Initialize perturbed state

2 **for** $i \leftarrow 1$ **to** $n$ **do**

3    $a'_t \leftarrow \pi(s'_t)$;      // Use learner policy to generate action

4    cost_grad $\leftarrow \nabla_{s'_t} c(\hat{f}(s'_t, \pi(s'_t)))$;      // Compute gradient

5    $s'_t \leftarrow s'_t + \kappa \cdot \epsilon \cdot \text{sign}(\text{cost\_grad})$;      // Gradient ascent

6    $s'_t \leftarrow \max(\min(s'_t, s_t + \epsilon), s_t - \epsilon)$;      // Clip perturbed state

7 **end**

8 $\delta_t \leftarrow s'_t - s_t$;          // Compute final perturbation

9 **return** $\delta_t$

---

### 4.2 ATTACK GENERATION

We leverage the learned constraint function $\psi$ and the learner policy $\pi$ to craft perturbations on the expert policy $\pi_E$. As defined in Section 3.3, our goal is to generate perturbations in the observation space that induce higher costs. Specifically, given the current observation $s_t$, we construct a perturbation $\delta_t$ such that the adversarial action $a'_t = \pi_E(s_t + \delta_t)$, when propagated through the estimated one-step system dynamics $\hat{f}$, drives the system to the next state $s'_{t+1} = \hat{f}(s_t, a'_t)$ that violates the learned safety constraints. Formally, the adversarial objective is defined as

$$\max_{|\delta_t| \leq \epsilon} \psi\big(f(s_t, \pi_E(s_t + \delta_t))\big), \tag{4}$$

where $\epsilon$ denotes the attack strength.

Next, to address this optimization problem, we employ a first-order approximation method to efficiently compute adversarial perturbations under norm constraints. Firstly, the next state $s_{t+1}$ is estimated through the function $\hat{f}$, obtained from a multi-layer perceptron (see Appendix C.1 for details). Then, to check whether $s_{t+1}$ violates the safety constraint, we use the cost function $\psi$ learned by ICRL to output the estimated cost value of $s_{t+1}$. Since our goal is to generate the perturbation $\delta_t$ to make the cost value as high as possible, we then use the gradient-based method to maximize the cost value. In the meantime, the perturbation is calculated by the gradient of the cost function and applied to the current state $s_t$. This process will be repeated until reaching the maximum number of iterations $n$. The final perturbation $\delta_t$ is computed as the difference between the original state $s_t$ and the perturbed state $s'_t$. The detailed attack generation process is shown in Algorithm 1.

### 4.3 THEORETICAL ANALYSIS OF ADVERSARIAL ATTACK

**Validity of the Learned Constraint Function.** Our attack framework builds upon the inverse constrained reinforcement learning (ICRL) paradigm proposed by Kim et al. (2024), in which the constraint function $\psi$ is recovered from expert demonstrations using a no-regret online learning algorithm (FTRL). According to Theorem 3.1 in (Kim et al., 2024), there exists an iterate $\psi$ such that the policy $\pi_L^{\psi}$ optimized under $\psi$ is $\xi$-approximately constrained-optimal, satisfying:

$$V_c^{\pi_L^{\psi}} - V_c^{\pi_E^{\psi}} \leq \xi \quad \text{and} \quad V_r^{\pi_L} \geq V_r^{\pi_E}. \tag{5}$$

This implies that the learned constraint function $\psi$ approximates the ground-truth constraint function $\psi^*$ asymptotically. As a result, optimizing under $\psi$ yields a policy that weakly Pareto-dominates the expert policy, meaning that the learned policy achieves at least as high a reward as the expert policy $\pi_E$ while satisfying the constraint approximately, up to a small violation $\xi$.

**Theorem 1 (Feasibility of Constraint-Based Perturbations).** *Let $\psi(s, a)$ be a constraint function learned via ICRL that satisfies $\forall(s, a)$, $|\psi(s, a) - \psi^*(s, a)| \leq \xi$, where $\psi^*$ denotes the ground-truth constraint. Let $\pi_E$ be the expert policy, and let $\delta$ be a perturbation such that $\psi(s + \delta, \pi_E(s + \delta)) > \eta$, where $\eta$ is the violation threshold. Then, if $\eta > \xi$, we have $\psi^*(s + \delta, \pi_E(s + \delta)) > \eta - \xi$, i.e., the perturbed state-action pair also violates the true constraint up to a bounded approximation error.*

Intuitively, Theorem 1 shows that adversarial perturbations based on the learned constraint effectively induce real constraint violations, provided the approximation error is small. The proof is provided in Appendix B.1.

**Lemma 1** (**Local Optimality of Gradient-Based Attacks**). *Let $\psi(s, a)$ denote the learned constraint function, and consider an adversarial perturbation $\delta$ on the state $s$ with magnitude bounded by $\epsilon$, i.e., $\|\delta\| \leq \epsilon$. Define the optimization problem of maximizing constraint violation as:*

$$\max_{\|\delta\| \leq \epsilon} \psi(s + \delta, \pi_E(s + \delta)).$$

*The perturbation generated by projected gradient ascent:*

$$\delta^* = \epsilon \cdot \frac{\nabla_s \psi(s, \pi_E(s))}{\|\nabla_s \psi(s, \pi_E(s))\|}$$

*is a locally optimal solution to this optimization problem.*

Lemma 1 ensures the local optimality of our gradient-based adversarial perturbations, validating our attack's theoretical effectiveness. The proof is provided in Appendix B.2.

**Lemma 2** (**One-Step Perturbation Cost Value Bound**). *Assume that the constraint function $\psi(s, a)$ is $L_\psi$-Lipschitz continuous with respect to state $s$, i.e., for all states $s$, actions $a$, and perturbations $\delta$, it holds that*

$$|\psi(s + \delta, a) - \psi(s, a)| \leq L_\psi \|\delta\|.$$

*Then, for any one-step perturbation $\delta$ with $\|\delta\| \leq \epsilon$, we have the following upper bound on the perturbed constraint value:*

$$\psi(s + \delta, \pi_E(s + \delta)) \leq \psi(s, \pi_E(s)) + L_\psi \epsilon.$$

Lemma 2 provides an explicit upper bound on the increase in constraint violation caused by a single-step perturbation. The proof is provided in Appendix B.3.

**Remark 1.** Compared to existing gradient-based adversarial attacks, which typically require explicit knowledge of the victim's policy or system dynamics to estimate theoretical bounds, our ICRL-based attack benefits from directly estimating the Lipschitz constant through the learned constraint network, thereby allowing **precise quantification** of attack strength and effectiveness.

**Lemma 3** (**Episodic Perturbation Cost Value Bound**). *Consider an episode of length $T$ with states $\{s_t\}_{t=1}^T$ generated by expert policy $\pi_E$. Assume that the constraint function $\psi(s, a)$ is $L_\psi$-Lipschitz continuous with respect to state $s$, and the system dynamics under $\pi_E$ and perturbations are Lipschitz continuous with constant $L_f$. If a perturbation $\delta_t$ with $\|\delta_t\| \leq \epsilon$ is applied at each step $t$, the cumulative constraint cost over the episode satisfies:*

$$\sum_{t=1}^T \psi(s_t + \delta_t, \pi_E(s_t + \delta_t)) \leq \sum_{t=1}^T \psi(s_t, \pi_E(s_t)) + \frac{L_\psi \epsilon (1 - (L_f)^T)}{1 - L_f}.$$

Lemma 3 extends this result by establishing a cumulative upper bound on constraint violation over an entire episode with sequential perturbations. The proof is provided in Appendix B.4.

In summary, this section provided theoretical validation, optimality analysis, and explicit bounds for our ICRL-based adversarial attack framework.

## 5 EXPERIMENTS

In this section, we present experiments to validate the effectiveness of our proposed method and address the challenges raised in Section 1. All experiments were conducted on a MacBook M4 Pro using Python 3.10.8, with Safe RL and attack algorithms implemented in PyTorch (Paszke et al., 2019). We evaluate our method on two safe RL tasks from Kim et al. (2024), which represent different levels of safety constraints: a **velocity-level constraint** (Safe-Ant-Velocity) and a **position-level constraint** (Safe-Ant-Position). Both expert agents were trained using PPO-Lagrangian Stooke et al.

(2020), with the cost limit set to 20 for Safe-Ant-Velocity and 100 for Safe-Ant-Position. The details of two environments are as follows.

**Safe-Ant-Velocity**: The agent is an Ant robot in PyBullet (Ellenberger, 2018–2019) simulator tasked with moving forward along the positive $x$-axis. Meanwhile, a velocity constraint is applied to restrict excessive or abrupt robot movements. The safety condition is defined as: $\|q_{t+1} - q_t\|_2 / dt \leq 0.75$, where $q_t$ denotes the position of the agent at time $t$.

**Safe-Ant-Position**: Similarly, a position constraint is imposed to the Ant agent to restrict unsafe movements. The safety condition is defined as: $0.5x_t - y_t \leq 0$, where $x_t, y_t$ denote the coordinates of the agent at time $t$.

For Safe-Ant-Velocity, the expert policy achieves an average reward of 1604.22 and an average cost of 16.26. For Safe-Ant-Position, the averages are 2066.06 and 77.90, respectively. These results indicate that the expert policies are well trained. Appendix C.2 provides additional training details and reports expert policies under different cost limits.

## 5.1 ICRL RESULTS

For Safe-Ant-Velocity, we train the ICRL algorithm for 20 epochs for the outer constrained RL policy learner and 10 epochs for the inner constraint learner. For Safe-Ant-Position, the training setup is similar, except that the inner constraint learner runs for 50 epochs. The results are illustrated in Figures 2. In both cases, the left plots show that the learned constraints progressively approach the ground-truth constraints over training epochs. The middle plots display the reward curves, indicating that the learner policies achieve even higher rewards than the expert policies. The right plots present the constraint violation curves, which decrease steadily over time. These results demonstrate that ICRL effectively learns both the safety constraints and high-performing policies from expert demonstrations in both environments.

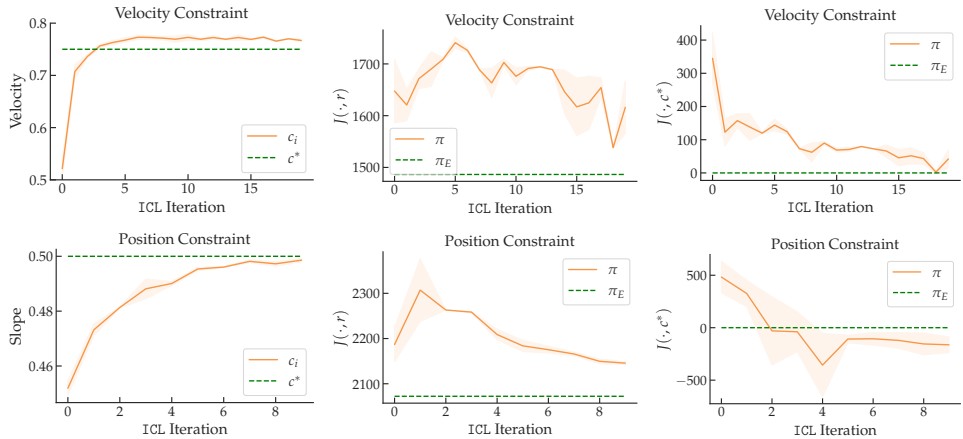

Figure 2: ICRL results for the Safe-Ant-Velocity and Safe-Ant-Position environments. The left figures show the constraint learning curve, the middle figures show the reward learning curve, and the right figures show the constraint violation curve.

## 5.2 ATTACK BASELINES

We compare our method with the following attack baselines:

**FGSM Attacker (L2):** This method (Goodfellow et al., 2014) uses the gradient of the target critic network, $grad = \nabla_s J(s, \pi^{target})$, along with uniform noise to iteratively perturb the state as $s_i = s - \text{sign}(grad) \cdot n_i$. If the perturbed state induces an undesirable action, it is stored as an adversarial sample.

**Gradient-based Attacker (L2):** This method (Pattanaik et al., 2017) uses the gradient of the target critic network, $grad = \nabla_s J(s, \pi^{target})$, along with uniform noise to iteratively perturb the state

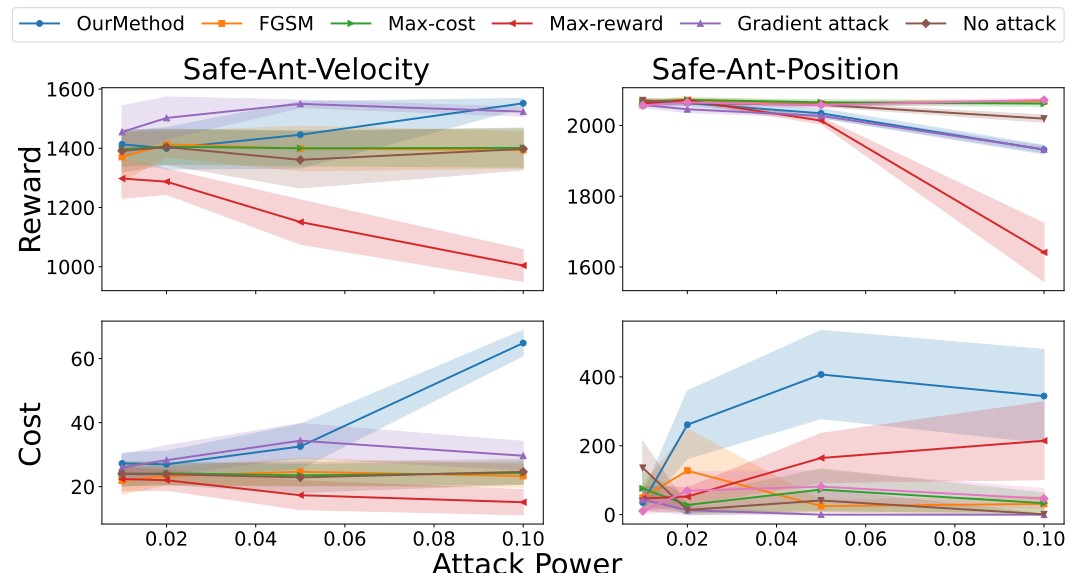

Figure 3: Adversarial attack results for Safe-Ant-Velocity and Safe-Ant-Position environments.

as $s_i = s - \text{sign}(grad) \cdot n_i$. If the perturbed state induces an undesirable action, it is stored as an adversarial sample.

**Max-Reward Attacker (L2):** This attacker (Liu et al., 2022b) is developed for safe RL tasks, which optimizes perturbations to maximize the reward critic of the victim policy to guide the policy violating the safety constraint. The loss function is defined as $l = -\|q_r\|$, where $q_r$ is the reward critic.

**Max-Cost Attacker (L2):** This attacker is also proposed in (Liu et al., 2022b), which maximizes the cost critic of the victim policy to guide the policy violating the safety constraint. The loss function is $l = -\|q_c\|$, where $q_c$ is the cost critic.

It is worth noting that **(L2)** defined in Table 1 means that the attack method requires access to the gradient information of the expert policy to generate perturbations, while our method only requires expert trajectories **(L1)**.

### 5.3 ATTACK RESULTS

**Adversarial Attack Performances** We evaluate adversarial attack performance in both Safe-Ant-Velocity (left two figures in Fig. 3) and Safe-Ant-Position (right two figures in Fig. 3). Each attack method is run over 50 episodes, and the reported curves show the average performance, with the shaded areas indicating standard deviations. We consider four perturbation strengths $\epsilon \in \{0.01, 0.02, 0.05, 0.1\}$.

In Safe-Ant-Velocity, the average reward across all attack methods remains within a similar range (around 1300–1600), indicating that reward is not affected. However, our method produces a substantially higher cost than all baselines. In particular, it reaches values above 60 at $\epsilon = 0.1$, whereas all other attack methods—including FGSM, max-cost, and max-reward—remain below 35 across all perturbation strengths. This shows that our method can effectively induce safety violations.

In Safe-Ant-Position, the distinction between methods is even clearer. While the average reward of our method is comparable to or slightly lower than baselines, its induced cost grows dramatically as $\epsilon$ increases. For instance, at $\epsilon = 0.05$, our method drives the cost beyond 400, far exceeding any baseline attack whose costs mostly remain below 130. In contrast, baseline methods such as FGSM and max-reward even achieve lower costs than the no-attack case, highlighting their ineffectiveness in destabilizing the policy.

To conclude, the results across both environments confirm that our method achieves significantly higher cost violations than all baseline approaches.

**Determining Attack Strength** As we stated in Section 1, a key challenge in adversarial attacks on Safe RL agents is to determine an appropriate attack strength that reliably induces safety violations without compromising stealth. If perturbations are too small, the induced constraint cost may be negligible, making the attack ineffective. Conversely, overly large perturbations risk being easily detected or unrealistic in practice.

As shown in Lemma 3, the increase in episodic cost is controlled by the Lipschitz continuity of the constraint function $\psi$ and the system dynamics, providing a principled link between the attack budget $\epsilon$ and the resulting cost violations. In practice, we evaluated reliability with 100 epochs of expert trajectories on Safe-Ant-Velocity. The estimated constants were $L_\psi \in \{0.0829, 0.1262, 0.1072, 0.1366\}$. Substituting these into Lemma 3, we obtained theoretical upper bounds $\{179.39, 231.58, 217.93, 238.39\}$, all of which safely exceeded the corresponding observed costs $\{42.08, 62.05, 141.05, 119.69\}$. This confirms that Lemma 3 provides valid and practical upper bounds. Conversely, when given a predefined cost value, we can calculate the required attack strength to induce that level of violation.

**Different Cost Limits** We further evaluate our method under cost limits of 0, 10, and 20 in the Safe-Ant-Velocity environment as an example, which are shown in Fig. 4. We can find from the figure that when the cost limit is 0, almost no violations are observed across all methods, and only small increases appear at $\epsilon = 0.05$, which is expected since the expert was trained to be strictly safe. With cost limit 10, our method begins to stand out, especially at $\epsilon = 0.05$, where it induces substantially higher costs than all baselines. For cost limit 20, our method consistently achieves the largest violations across all perturbation levels, and the gap with baselines becomes more pronounced. Overall, these results show that our method generalizes well across different expert policies and consistently induces stronger safety violations.

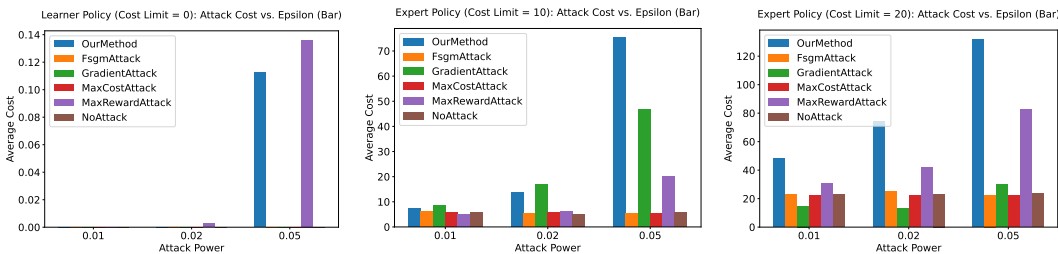

Figure 4: Adversarial attack results under different cost limits in the Safe-Ant-Velocity environment. Left: cost limit = 0. Middle: cost limit = 10. Right: cost limit = 20.

# 6 DISCUSSION

In this work, we proposed an adversarial attack framework that addresses key challenges in analyzing vulnerabilities of safe reinforcement learning (Safe RL) policies. First, our method requires only expert demonstrations to conduct attacks, making it more practical and less detectable. Second, by leveraging the safety constraint network and learner policy obtained via ICRL, it can estimate the expected cost bound and set the attack budget accordingly. Finally, the framework provides a systematic way to study the vulnerabilities and robustness of Safe RL policies, offering new insights for developing safer learning systems.

**Limitations:** One limitation of our method is that when demonstrations are unrepresentative or lack safety information, the ICRL process may fail to recover the safety constraint function, thereby reducing attack effectiveness.

**Defense:** Our method exposes the vulnerability of safe RL policies to adversarial attacks. Equally important is the need to enhance the robustness of these policies against such attacks. One effective strategy is adversarial training, which incorporates adversarial perturbations into the training process to improve the policy's resilience. Another promising approach is probabilistic shielding (Belardinelli et al., 2025), which identifies and prevents actions that are likely to violate safety constraints. Furthermore, as discussed in the limitations, explicitly encoding safety constraints in the state representation can significantly reduce the success rate of our attack.

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

# A    USE OF LARGE LANGUAGE MODELS (LLMS)

This work used large language models (LLMs) in the following ways:

- **Purpose**: grammar and wording checking and improvements.
- **Scope**: The LLM was only used for improving clarity of writing, and all scientific claims, analyses, and conclusions were made and verified by the authors.
- **Responsibility**: The authors take full responsibility for the correctness and integrity of all contents, including any parts where LLMs were used for assistance.

# B    PROOF OF THEOREMS

## B.1    PROOF OF THEOREM 1

*Proof.* By the uniform approximation bound, we have

$$|\psi(s,a) - \psi^*(s,a)| \leq \epsilon \quad \Rightarrow \quad \psi^*(s,a) \geq \psi(s,a) - \epsilon.$$

Substituting $(s + \delta_s, \pi_E(s + \delta_s))$ into the above, we get

$$\psi^*(s + \delta_s, \pi_E(s + \delta_s)) \geq \psi(s + \delta_s, \pi_E(s + \delta_s)) - \epsilon > \delta - \epsilon.$$

Therefore, the perturbed state-action pair results in a violation of the true constraint when $\delta > \epsilon$.   $\square$

## B.2    PROOF OF LEMMA 1

*Proof.* We prove local optimality by contradiction. Assume, for the sake of contradiction, that there exists another perturbation $\hat{\delta}$ with $\|\hat{\delta}\| \leq \epsilon$ in a local neighborhood of $s$, such that:

$$\psi(s + \hat{\delta}, \pi_E(s + \hat{\delta})) > \psi(s + \delta^*, \pi_E(s + \delta^*)).$$

Since $\psi(s,a)$ is differentiable with respect to $s$, we have the first-order Taylor expansion around the point $(s, \pi_E(s))$:

$$\psi(s + \hat{\delta}, \pi_E(s + \hat{\delta})) \approx \psi(s, \pi_E(s)) + \nabla_s \psi(s, \pi_E(s))^\top \hat{\delta}.$$

The perturbation $\delta^*$ defined by projected gradient ascent maximizes the linear approximation:

$$\delta^* = \arg \max_{\|\delta\| \leq \epsilon} \nabla_s \psi(s, \pi_E(s))^\top \delta.$$

By the definition of the maximization, we have:

$$\nabla_s \psi(s, \pi_E(s))^\top \hat{\delta} \leq \nabla_s \psi(s, \pi_E(s))^\top \delta^*.$$

This implies:

$$\psi(s + \hat{\delta}, \pi_E(s + \hat{\delta})) \leq \psi(s, \pi_E(s)) + \nabla_s \psi(s, \pi_E(s))^\top \delta^*.$$

But from the Taylor approximation for $\delta^*$, we also have:

$$\psi(s + \delta^*, \pi_E(s + \delta^*)) \approx \psi(s, \pi_E(s)) + \nabla_s \psi(s, \pi_E(s))^\top \delta^*.$$

Therefore, we must have:

$$\psi(s + \hat{\delta}, \pi_E(s + \hat{\delta})) \leq \psi(s + \delta^*, \pi_E(s + \delta^*)),$$

which directly contradicts our initial assumption. Thus, there is no perturbation in a local neighborhood around $s$ that achieves higher constraint violation than the gradient-based perturbation $\delta^*$. Hence, the gradient-based perturbation is locally optimal.   $\square$

### B.3 PROOF OF LEMMA 2

*Proof.* By definition of Lipschitz continuity, we have:

$$|\psi(s + \delta, a) - \psi(s, a)| \leq L_\psi \|\delta\|.$$

Substituting $a = \pi_E(s + \delta)$, and noting $\|\delta\| \leq \epsilon$, we immediately obtain:

$$\psi(s + \delta, \pi_E(s + \delta)) \leq \psi(s, \pi_E(s)) + L_\psi \epsilon.$$

Further noting that policy $\pi_E$ is stationary and locally continuous, we have that, in a sufficiently small local neighborhood,

$$|\psi(s, \pi_E(s + \delta)) - \psi(s, \pi_E(s))| \approx 0,$$

thus arriving at the simpler bound:

$$\psi(s + \delta, \pi_E(s + \delta)) \leq \psi(s, \pi_E(s)) + L_\psi \epsilon.$$

This completes the proof. □

### B.4 PROOF OF LEMMA 3

*Proof.* We start from the Lipschitz continuity assumption:

$$|\psi(s_t + \delta_t, \pi_E(s_t + \delta_t)) - \psi(s_t, \pi_E(s_t))| \leq L_\psi \|s_t + \delta_t - s_t\| = L_\psi \|\delta_t\| \leq L_\psi \epsilon.$$

Thus, at each step $t$, we have the following bound:

$$\psi(s_t + \delta_t, \pi_E(s_t + \delta_t)) \leq \psi(s_t, \pi_E(s_t)) + L_\psi \epsilon.$$

Now, note that the perturbation at each time-step affects subsequent states through system dynamics. Assuming the system dynamics are Lipschitz continuous with constant $L_f$, we have:

$$\|s_{t+1} - \hat{s}_{t+1}\| \leq L_f \|s_t + \delta_t - s_t\| \leq L_f \epsilon,$$

where $\hat{s}_{t+1}$ denotes the state without perturbation at time $t + 1$.

Iteratively applying this logic, at time step $t$, the perturbation effect accumulates as:

$$\|s_t + \delta_t - s_t\| \leq L_f^{t-1} \epsilon.$$

Thus, the episodic cumulative constraint cost under perturbation satisfies:

$$\sum_{t=1}^{T} \psi(s_t + \delta_t, \pi_E(s_t + \delta_t)) \leq \sum_{t=1}^{T} \psi(s_t, \pi_E(s_t)) + L_\psi \epsilon \sum_{t=1}^{T} L_f^{t-1}.$$

The geometric series on the right side can be simplified as:

$$\sum_{t=1}^{T} L_f^{t-1} = \frac{1 - L_f^T}{1 - L_f}.$$

Thus, we have the final bound:

$$\sum_{t=1}^{T} \psi(s_t + \delta_t, \pi_E(s_t + \delta_t)) \leq \sum_{t=1}^{T} \psi(s_t, \pi_E(s_t)) + \frac{L_\psi \epsilon (1 - L_f^T)}{1 - L_f}.$$

This concludes the proof. □

## C  Implementation Details

### C.1  System Identification

To estimate the next state $s_{t+1}$ and compute state gradients, we perform system identification by training a multi-layer perceptron (MLP) on expert trajectories. Given $(s_t, a_t, s_{t+1})$ tuples collected from the expert policy $\pi_E$, the MLP learns the mapping

$$s_{t+1} = \hat{f}(s_t, a_t). \tag{6}$$

This learned model $\hat{f}$ serves two purposes: (i) predicting the next state $s_{t+1}$ for adversarial attack generation, and (ii) providing a differentiable approximation that allows computing $\nabla_{s'_t} c(\hat{f}(s'_t, \pi(s'_t)))$ in Algorithm 1.

Taking the Safe-Ant-Velocity environment as an example, we collect 100 epochs of trajectories ($100,000$ steps) under the expert policy. The MLP is trained on these data and evaluated on 50 unseen trajectories, achieving a mean squared error (MSE) of $0.000672$, which demonstrates high predictive accuracy. Despite being approximate, the learned dynamics captures local transition behavior effectively, enabling reliable gradient-based perturbation generation without assuming access to the true dynamics.

### C.2  Expert Policy Training

To establish reliable expert policies, we trained the Safe-Ant-Velocity agent under three different cost limits: 0, 10, and 20. The evaluation results are summarized in Table A.1. Each value represents the average reward and cost over 100 evaluation epochs.

Table 2: Training results of Safe-Ant-Velocity expert policies under different cost limits.

| Cost Limit | Average Reward | Average Cost |
|:---:|:---:|:---:|
| 0 | 1505.90 | 0.00 |
| 10 | 1557.99 | 2.94 |
| 20 | 1604.22 | 16.26 |

As shown in Table 2, the expert policies achieve consistently high rewards while keeping the average cost within the designated limit. These policies are subsequently used as baselines for evaluating learner policies and adversarial attack methods in the main text.

### C.3  Experimental Environment and Runtime

All experiments were conducted on a MacBook M2 Pro with 16 GB RAM, using Python 3.10.8. The memory footprint remained well within this limit during both training and attack phases. Following the reviewer's suggestion, we additionally measured the wall-clock runtime of data collection and attack execution.

**Trajectory collection:**

- In the Ant-Velocity environment, collecting 20 episode trajectories on CPU (4 threads) required 51.9 s ($\approx$2.59 s/episode).
- In the Ant-Position environment, the collection took 50.88 s ($\approx$2.54 s/episode).

**Attack runtime:**

- We evaluated Algorithm 1 by running 50 episodes under four attack strengths $0.01, 0.02, 0.05, 0.1$, using multiprocessing with four CPU cores.
- The total average runtime across all strengths was 23.24 s/episode, which corresponds to $\approx$ 5.81 s/episode for a fixed attack strength.
- Under the same settings, Safety-Ant-Velocity required 4.71 s/episode, while Safety-Ant-Position required 6.06 s/episode.

These results confirm that the proposed attack generation method is computationally feasible on standard hardware without GPU acceleration.

