# OpenReview forum: "Vulnerability Analysis of Safe Reinforcement Learning via Inverse Constrained Reinforcement Learning"
_ICLR.cc/2026/Conference — Submitted to ICLR 2026_

### Official Review · Reviewer_ofMg · 2025-10-28

**Soundness:** 2
**Presentation:** 2
**Contribution:** 2
**Rating:** 2
**Confidence:** 4

**Summary:**

This paper proposes a state-perturbation attack method to increase the safety cost of an RL agent subject to a safety constraint. The approach first applies the inverse constrained reinforcement learning (ICRL) method recently proposed in [Kim et al., 2024] to learn the cost function from a set of clean trajectories and the agent's policy and reward function. It then applies the standard projected gradient descent (PGD) method to identify state perturbations that maximize the agent's instantaneous cost in each time step.

**Strengths:**

The paper shows that it is possible to perturb an agent's states to increase its safety cost even without knowing the cost function a priori, as long as the attacker has access to a set of clean trajectories and the agent's policy and reward functions.

**Weaknesses:**

1. The paper applies the ICRL method in [Kim et al., 2024] to estimate the agent's cost function and then leverages the standard PGD attack. The technical contribution is low.
2. While the paper claims its approach to be gradient-free, it looks like it still needs the gradient information. In Algorithm 1, line 4, the PGD method requires evaluating the gradient with respect to s', which involves the gradient of pi_E.
3. The attacker's objective is confusing. According to (4), it appears the attacker simply maximizes the instantaneous cost at each time step, which is inconsistent with the cumulative cost constraint in the safe RL formulation (1). Further, in this case, the attacker's problem is essentially the same as the traditional reward minimization problem considered in the literature for unconstrained MDPs.
4. The evaluation is unfair by comparing the proposed method with baselines developed for different objectives other than (4). Given the myopic objective (4), PGD is the most direct approach to optimize it. It makes better sense to just compare the same PGD attack with and without ICRL.
5. Another big issue with the evaluation is that it completely ignores defenses. As the attacker in the paper only considers the cost objective, previous defenses for unconstrained MDPs can be easily adapted to the scenario considered in this paper.
6. The paper has multiple typos and missing/inconsistent definitions.
 - The constraint function phi is undefined. It looks like phi should just be the cost function c in the safe MDP formulation, although that is never made clear in the paper.
 - Similarly, it looks like the so-called system identification function f is simply the transition dynamics p.
 - The definition of stealthness is not standard. In adversarial machine learning, one typically captures stealthiness based on the attack outcome (intensity, frequency, etc.), not the attacker's capability (black-box, white-box, etc.).
 - In the safe MDP definition, the cost function c relies on the current state, the current action, and the next state, but in Algorithm 1 and Eq. (4), it only depends on the next state. Then, in Lemma 1, it only depends on the current perturbed state and action, which is very confusing.
7. The related work section misses some important recent studies on state perturbations against RL, such as
- Li et al., Towards Optimal Adversarial Robust Q-learning with Bellman Infinity-error, ICML 2024.
- Sun et al., Belief-Enriched Pessimistic Q-Learning against Adversarial State Perturbations, ICLR 2024.
- Yang et al., DMBP: Diffusion Model-Based Predictor for Robust Offline Reinforcement Learning against State Observation Perturbations, ICLR 2024.
- Liu et al., Beyond Worst-case Attacks: Robust RL with Adaptive Defense via Non-dominated Policies, ICLR 2024.

**Questions:**

Please refer to the discussion on weaknesses above.

---

### Official Review · Reviewer_yhh7 · 2025-10-29

**Soundness:** 2
**Presentation:** 3
**Contribution:** 2
**Rating:** 4
**Confidence:** 3

**Summary:**

This paper presents a novel framework for analyzing the vulnerabilities of Safe Reinforcement Learning (Safe RL) agents using inverse constrained reinforcement learning (ICRL). The authors show that even safety-constrained policies remain susceptible to adversarial manipulation without requiring gradient access to the victim model. By reconstructing implicit constraints from expert demonstrations, the proposed approach systematically reveals hidden weaknesses in Safe RL algorithms, offering both theoretical insights and empirical validation across benchmark environments.

**Strengths:**

1. Introducing an inverse-learning formulation to analyze Safe RL vulnerabilities is original and conceptually strong.

2. The method works under limited access assumptions, making it applicable to realistic black-box attack settings.

3. This work provides a theoretical analysis of the attack performance bound, which strengthens the work's contribution.

**Weaknesses:**

1. The methodology is relatively trivial by combining ICRL to learn the constraint policy and an approximate agent policy to conduct gradient-based attack.

2. The experiment was only conducted on two toy environments, which is not general enough to state the effectiveness of the proposed method. I would like to see results on more environments.

3. The experiment only compares with other attack methods. It does not include defense baselines in the experiment.

**Questions:**

1. The proposed method will rely on how accurately the approximate safety constraint function and learner policy are learned through expert trajectories. Could the author provide analysis or ablation studies on how many trajectories are needed?

2. For the question refer to the experiment, please refer the Weakness 2 and 3.

3. Will the method scale to an image-based RL agent?

---

### Official Review · Reviewer_jVQr · 2025-10-31

**Soundness:** 2
**Presentation:** 2
**Contribution:** 2
**Rating:** 2
**Confidence:** 4

**Summary:**

This paper addresses the vulnerability assessment of Safe RL in Constrained Markov Decision Processes. existing Safe RL methods often assume benign environments, limiting their real-world applicability. To address this, they propose a method to evaluate the maximum cost under realistic adversarial attacks. In their threat model, the adversary has access to the victim’s demonstrations but not to the policy gradients and constraint functions. Under this setting, the authors propose a method that combines Inverse Constrained Reinforcement Learning (ICRL) with gradient-based adversarial perturbations. Specifically, they estimate a differentiable constraint function via ICRL and optimize perturbations to maximize this estimated constraint. Experiments show that their method achieves higher attack performance than attacks assuming even stronger adversarial capabilities.

**Strengths:**

The paper provides a practical and realistic vulnerability assessment of Safe RL by considering adversaries closer to real-world scenarios. This perspective enhances the practical value of robustness evaluation for Safe RL methods.

**Weaknesses:**

The novelty of the proposed method is limited. It is essentially a straightforward combination of ICRL and standard gradient-based adversarial optimization. The theoretical analysis in Section 4.3 also lacks originality. Theorem 1 can be almost trivially derived from [1], and Lemmas 2, 3, and 4 follow directly from their assumptions. Although Remark 1 claims the effectiveness of estimating the Lipschitz constant, this claim is not empirically validated.

Moreover, the proposed method is computationally expensive. It requires estimating both the cost function via ICRL and the system dynamics surrogate $\hat{f}$. These estimations require a very large computational cost. Consequently, its applicability to high-dimensional state or action spaces may be severely limited.

Minor comment:
The descriptions of FGSM Attacker (L2) and Gradient-based Attacker (L2) appear almost identical except for the citations. Clarifying the difference would strengthen the experimental section.

[1] Konwoo Kim, Gokul Swamy, Zuxin Liu, Ding Zhao, Sanjiban Choudhury, and Steven Z Wu. Learning shared safety constraints from multi-task demonstrations. Advances in Neural Information Processing Systems, 36, 2024.

**Questions:**

- Why does the proposed method outperform the Max-Cost Attacker, which directly uses the cost critic and should act as an oracle?
- How computationally expensive is the cost estimation process in the proposed method, and is it feasible for high-dimensional environments?
- To what extent does the quality of the system dynamics surrogate $\hat{f}$ affect attack performance, and how costly is this estimation?
- The experiments use 100 epochs of expert trajectories, which is relatively large. How does the number of demonstrations correlate with attack performance?
- Equation (4) appears to require the gradient of $\pi_E$, but this is not permitted under the threat model. Is this a typo?

---

### Meta-Review · Area_Chair_XQuW · 2026-01-02

**Summary:**

All three reviewers tend to reject this paper, with the scores of 2, 4, 2 (2.67 on average). I find several major concerns from the reviewers:
(1) The novelty is limited. It seems like a straightforward combination of ICRL and standard adversarial optimization and does not demonstrate very promising results. (All three reviewers point out this weakness).
(2) The method is computationally expensive. The experiment was only conducted on two toy experiments.
(3) Many details are unclear, such as the attacker's objective, evaluation fairness, the constraint function phi, and the definition of stealthness.
(4) Insufficient discussion on related works, such as [1]. One reviewer stated that Theorem 1 of this paper can be almost trivially derived from the reference [1], and Lemmas 2, 3, and 4 follow directly from its assumptions.

[1] Konwoo Kim, Gokul Swamy, Zuxin Liu, Ding Zhao, Sanjiban Choudhury, and Steven Z Wu. Learning shared safety constraints from multi-task demonstrations. Advances in Neural Information Processing Systems, 36, 2024.

**Reviewer Concerns:**

The authors did not provide rebuttal, and all concerns remain

**Reviewer Scores:**

All reviewers will keep scores and tend to reject

---

### Decision · Program_Chairs · 2026-01-26

Reject